# *RootLogChain*: Registering Log-Events in a Blockchain for Audit Issues from the Creation of the Root

**DOI:** 10.3390/s21227669

**Published:** 2021-11-18

**Authors:** Juan Carlos López-Pimentel, Luis Alberto Morales-Rosales, Raúl Monroy

**Affiliations:** 1Facultad de Ingeniería, Universidad Panamericana, Álvaro del Portillo 49, Zapopan 45010, Jalisco, Mexico; 2Facultad de Ingeniería Civil, CONACYT-Universidad Michoacana de San Nicolás de Hidalgo, Morelia 58000, Michoacán, Mexico; lamorales@conacyt.mx; 3School of Engineering and Sciences, Tecnologico de Monterrey, Av. Lago de Guadalupe Km 3.5, Atizapán de Zaragoza 52926, Edo. Mexico, Mexico; raulm@tec.mx

**Keywords:** blockchain, audit, root creation, security protocols, smart contracts

## Abstract

Logging system activities are required to provide credibility and confidence in the systems used by an organization. Logs in computer systems must be secured from the root user so that they are true and fair. This paper introduces *RootLogChain*, a blockchain-based audit mechanism that is built upon a security protocol to create both a root user in a blockchain network and the first log; from there, all root events are stored as logs within a standard blockchain mechanism. *RootLogChain* provides security constructs so as to be deployed in a distributed context over a hostile environment, such as the internet. We have developed a prototype based on a microservice architecture, validating it by executing different stress proofs in two scenarios: one with compliant agents and the other without. In such scenarios, several compliant and non-compliant agents try to become a root and register the events within the blockchain. Non-compliant agents simulate eavesdropper entities that do not follow the rules of the protocol. Our experiments show that the mechanism guarantees the creation of one and only one root user, integrity, and authenticity of the transactions; it also stores all events generated by the root within a blockchain. In addition, for audit issues, the traceability of the transaction logs can be consulted by the root.

## 1. Introduction

Auditing is an independent and objective activity to add value to an organization, evaluating the efficiency or efficacy of its processes. A key requirement of auditing is to provide data integrity throughout the system’s life cycle to be audited since stored data can be vulnerable to manipulation in good or bad faith, especially when the data are in the cloud. Data integrity mechanisms aim to protect information against modification, be it by an intruder, software malfunctioning, or simply user negligence.

Implementing strict auditing is challenging on any distributed system, particularly for servers deployed in a hostile environment. For auditing, things are not different: if logs are not secured, the history of events can be altered [1]. Audit logs are used to keep track of important events about system activities and are a fundamental mechanism for digital forensics because they provide information about past and current events and hence, the path of states of a system [2]. The need for protecting logs from attackers was already stated by various researchers in different contexts, in the context of hardware [3]; systems [4,5]; file systems [6]; databases [7]; and secure logging protocols [8]. Companies are currently attracted to migrate to cloud computing services [9]. Although cloud computing opens a new horizon of computing for IT organizations [10], it also opens more opportunities for criminal activity. Current research has focused on audit logging in distributed systems [2,10,11,12,13,14]. It is known that distributed systems work on the internet platform, which was built as a means to guarantee communication, not to ensure information security properties. That is why most solutions deployed over the internet notably raise concerns in terms of trust and privacy.

Blockchain has emerged as a native technology that works strongly in terms of trust for critical applications, such as logging applications. This technology has transformed the way data are stored, giving them greater security and better quality and confidence. Chowdhury et al. presented an analysis, reporting that blockchain can be applied to more critical scenarios [15]. Due to the transparency offered by blockchain technology, there is an increasing number of new mechanisms that use this technology for audit logging [1,16,17,18,19,20]. We have found that these solutions, however, do not show enough detail about how the transactions are formed through smart contracts. In addition, we have identified that blockchain solutions on audit logging have not focused on detailing aspects about how the phase of creating the first user in a system, so-called *the root*, is created and how these creation events are recorded in logs. The *root* typically has access rights to create other users. The creation of the *root* account is critical, especially nowadays, where the trend is the development of cloud applications. Unlike former times, it is very common to remotely create the first user of a distributed system. Yet, in the middle of the network (i.e., the internet), an attacker, for example, an eavesdropping attacker, could be present and take advantage of a root creation action to then launch a data integrity attack. Hence, we must track and store all the activities in a reliable technology to first guarantee a reliable start and then further traceability for auditing issues.

The main objective of this research is to introduce *RootLogChain*, a secure mechanism for the initialization phase of a system, from which we can establish that logs are secure, even from root creation. This initialization phase consists of two stages: in the first stage, a security protocol is run to permit the creation of the *root*, regardless of it being in a hostile distributed system; the root creation is stored in a blockchain through an *Audit Server*. Blockchain is used to guarantee the integrity of the events that are stored using smart contracts. In the second stage, our mechanism catches all events generated by the root, and stores them in the smart contract generated in the first stage. With such a smart contract, we can ask queries regarding any events that are logged. This provides data integrity, guaranteed by the strength of the blockchain.

We have implemented *RootLogChain* in a prototype. It works in a distributed context, using a microservice architecture, implementing the Ethereum infrastructure on the blockchain side. We have developed various security experiments in our setting. We have verified from our implementation that our mechanism allows the creation of a unique root, and that the log events are stored in a blockchain from the first event that the root was created and for subsequent events.

We summarize the main contributions of this work as follows:Ensure a secure mechanism for creating the root (first and unique user) in a distributed environment that allows to record and secure the transactions generated with smart contracts using blockchain.Provide a mechanism for auditing issues in the initialization phase of a system guaranteed by blockchain’s strengths concerning integrity and authenticity.Perform traceability of each of the transactions carried out by the root through secure logs in distributed environments.

The paper is organized as follows: Section 2 gives an overview of some of the technologies above mentioned and outlines related work. Section 3 describes the methodology, the general scenario and describes the main system requirements. Section 4 explains in detail root creation through the use of a security protocol. Section 5 explains how events of the root are stored in the blockchain. The details about how the smart contracts work within the blockchain are explained in Section 6. Section 7 describes a prototype. The experiments and validations of our mechanism are reported in Section 8. Finally, conclusions drawn from our research are summarized in Section 9.

## 2. Background and Related Work

This section is split into three sections. In the first one, we argue the crucial nature of the correct design of a security protocol; then, in the second subsection, we provide an overview of proposals for implementing an audit system; in the last subsection, we elaborate on auditing issues but in a more specialized area—the blockchain.

### 2.1. Security Protocols

When developing distributed applications, we must consider that active and passive attackers are always present on the network; attackers are able to see and manipulate any message sent over the network. Accordingly, any non-encrypted message transmitted on the internet, for example, is considered unsafe. Therefore, if we want to provide a security guarantee, it is necessary to use security mechanisms, including security protocols.

A security protocol, also known as a cryptographic protocol, is a set of rules and conventions whereby one or more agents agree about each others’ identity, usually ending up in possession of one or more secrets [21]. Each protocol is defined in two parts: the first states initial knowledge of the participants, and the second enumerates a set of steps, where participants exchange messages. Apart from sending or receiving a message, a participant can perform some computations (which can be dependent on the previously received message).

Security protocols are used to provide a wide variety of security services, including key distribution, data confidentiality, authentication, and non-repudiation [22]. Although security protocols consist of only a few messages, their design is amazingly error prone. This explains the interest of the formal methods community in providing mechanical proofs of their properties [23,24,25]. Recently, this community has also given attention to justifying the formal verification on blockchain systems by the added security [26], focusing, in particular, on smart contracts [27].

### 2.2. Audit Based Systems

Audit logs are used to keep track of events of interest about system activity, such as program execution/computer crash, data modification, and user activities. They are a fundamental digital forensic mechanism to provide information about the current and past states of a system [2]. Audit logs are themselves a key target for an attacker who needs to erase any trace of their malicious activities; otherwise, they may get caught and then possibly prosecuted. The need for securing audit logging was raised already in different contexts, including hardware [3]; systems [4,5]; file systems [6]; databases [7]; secure logging protocols [8]; distributed systems [2,10,11,12,13,14]; blockchain [1,16,17,18,19,20]; and blockchain hardware [28], as well as many others. In this section, we describe some pieces of research in securing audit logging, with particular attention on distributed systems and cloud computing; we describe blockchain-based mechanisms later in the following subsection.

Yavuz and Ning proposed a logging scheme for distributed systems, called *blind-aggregate-forward* (BAF) [2]. It signs a log entry, using public-key cryptography. BAF aims to provide low logger computational overhead, near-zero storage and communication overheads, public verifiability, and immediate log verification. Since secure logging mechanisms based on public-key cryptography involve expensive operations, Yavuz et al. proposed LogFAS [11]; LogFAS combines symmetric and asymmetric cryptography, yielding a more efficient mechanism, both in terms of time and storage.

Nowadays, companies are attracted to migrating to a cloud computing environment. This is because it offers cost benefits, as it does not require any local infrastructure setup. Yet, cloud computing also comes with some risks [9,10], namely, it can be easily exploited by attackers who run malicious code on machines inside that cloud. Cloud forensics poses strong challenges to cloud service providers. For example, due to the decentralized nature of data processing in the cloud, traditional approaches that gather evidence or provide recovery services are no longer practical [29]. Accordingly, Zawoad et al. [10] proposed a mechanism called *SecLaaS*, which stores virtual machines’ logs and provides access to forensic investigators, ensuring the confidentiality of cloud users.

Preserving the confidentiality and integrity of logs in distributed systems and cloud services is also a key problem. To deal with this issue, mechanisms based on blockchain were recently put forward.

Blockchain combines hashing and cryptography, along with a decentralized structure, making it extremely difficult for any third party to manipulate information. Chowdhury et al. [15] presented a comparative study of standard databases and those based on blockchain technology. They developed a decision tree diagram that aims to help practitioners and researchers choose the appropriate technology, depending on the application. Their analysis concluded that blockchain can be applied to deal with more critical scenarios. Blockchain technology has proven helpful in guaranteeing reliability on critical platforms, such as audit logging and digital finance; however, its potential is enormous in almost any area in which information transactions are involved [30,31,32].

### 2.3. Blockchain in Auditing

Blockchain technology, introduced by Satoshi Nakamoto [33], is described as a distributed data structure, where the information contained is allocated in blocks establishing a chain. Each block in the chain contains the hash address of its predecessor and one or more transactions, among other data. Blockchain is resistant to malicious data modification; its distributed ledger design can record transactions between two or more parties in a verifiable and permanent way. In the beginning, blockchain technology was developed to guarantee integrity in crypto coins. However, it was recently used to provide other security properties for different applications, such as data management, financial services, cybersecurity, IoT, food science, healthcare industry, and brain research [30]. It is also applied in the context of vehicles [34,35], e-voting [36] and Industry 4.0 [37].

Blockchain technology likely poses one of the most significant developments in information technology in recent years. For example, Makridakis et al. [31] provided evidence that blockchain is remarkable, and they argued that the importance of blockchain can be evidenced by the number of papers published on the topic. They also argued that it will change the way transactions are made. Their stance is based on blockchain’s ability to guarantee trust among unknown actors and ensure the immutability of records while also making intermediaries obsolete.

A blockchain can also hold programmed instructions, so-called *smart contracts*. Smart contracts reside at a specific address on the blockchain, called *smart contract address*; they are run by a miner and validated by all network miners. If anyone wants to change something, it must be done via a smart contract, and the transaction has to be accepted by all miners. Transactions are never deleted from a chain, and so any change can be verified by all miners. Because of that, it is an appropriate technology for auditing.

Shigeya Suzuki et al. [1], for instance, proposed a blockchain mechanism used as a request–response channel for a client–server system. There, the blockchain is used to record transactions requested by clients and replied by the server. Both the clients and server must continuously watch for any transactions sent to them. The duration of a transaction, while the blockchain is acting as a channel, is one of their disadvantages, as the chain becomes extremely slow.

Ahmad et al. [16] used a Hyperledger blockchain and proposed a system called *BlockAudit*, which enables a system to avoid audit logs from being tampered with by adversaries. More recently, Ahmad et al. [17] provided a detailed account of how BlockAudit works. In their scenario, they use a web application to generate logs, which are then stored in a traditional database. Then, logs are sent to the blockchain in a format based on JavaScript Object Notation (JSON). Each JSON packet is sent to the blockchain network as a new transaction.

Putz et al. [18] proposed an architecture for log integrity preservation. It uses blockchain to store log records generated by different sources, such as containerized applications, firewalls, or intrusion detection systems. They introduced a prototype based on a SIEM reference architecture, whose inputs were collected from a web application. The server part was implemented with Node.js and the Exonum blockchain. Their experiments showed high storage cost, and they also did not detail how they implemented the transactions within the smart contracts on the blockchain.

We have identified that previous research does not detail aspects of the initial phase when a first user in a system is created and how these first events (in log presentation) are recorded in the blockchain. In addition, we have identified that these solutions do not show enough detail about how transactions are constituted through smart contracts.

## 3. The General Design Model

Our research consists of an application of design science research in the information systems area. In particular, we have followed the methodology proposed by Peffers et al. [38], which is commonly applied in engineering (and so it should be applicable to software engineering).

The methodology is composed of the following steps: (a) problem identification and motivation, (b) definition of the objectives for a solution, (c) design and development, (d) demonstration, and (e) evaluation. We have already dealt with steps (a) and (b) (see Section 1 and Section 2.) In this section, we explain the general design of our method, providing a detailed elaboration below (see Section 4, Section 5 and Section 6). We outline our prototype in Section 7 and provide a summary of the results yielded from its validation in Section 8.

### 3.1. A General Design of Users Creation in Distributed Systems

Traditionally, a root user in an operating system is the account that has access to all files and to all commands (hence, it has access rights to create other user accounts). It runs under other names, including superuser, or administrator (often called admin, for short). Henceforth, we shall refer to it as the *root user*, or simply the *root*. We now introduce *RootLogChain*, which is capable of creating the root in a distributed environment and recording and securing any transaction via smart contracts in a blockchain. Figure 1 depicts our working scenario, which has two phases: initialization and deployment. *RootLogChain* is concerned with the initialization phase only. Each phase contains one or more steps, denoting a number enclosed by a solid circle. Overall, the workflow is as follows:**Initialization** **phase**: This phase comprises all events related to root creation and those derived from root executing a number of commands, following its creation. A distributed system could be composed of different client–server systems, each of which may be configured with traditional or federated databases. Any of these systems must have their own root user. In this initialization phase, anyone who is to become the root must run a protocol with a remote server (see step 1 in Figure 1). Depending on the system, root privileges can be unlimited or limited to creating other types of users (administrators with fewer privileges or simply operator users). After its creation, in step 2, the root creates one or more users (three, in this case).**Deployment** **phase**: This phase abstracts out a number of transactions executed by different users in a system, including the creation of other users by an administrator (see step 3 in Figure 1). Events caused by other users are not dealt with *RootLogChain*, but by other solutions as explained in [39,40].

### 3.2. Internal Workings of the Initialization Phase

There are two key problems with remotely creating the root in the context of a client–server model or a more complex distributed model. One is that an intruder might impersonate any party involved in the process, and the other is that an intruder may alter the audit logs used to convey root creation and any subsequent activity. Figure 2 depicts our architecture protocol. It has two parts: the left-hand side part (the architecture itself, explained in detailed in the next subsection) and the right-hand one, the protocol, where four entities—agent A, *Root*, Audit Server (abbreviated as AS) and the Blockchain—are engaged in a two-stage procedure. The stages are as follows: **Root****creation:** For this task, we propose a security protocol. Any user A wanting to be a root plays a run of the protocol with AS; the result is the creation of the root, shown in Figure 2 with a down arrow showing the transition of any user becoming the root. In this stage, smart contracts are created in the blockchain and the logs corresponding to such events are also stored. **Root****events:** Once a user is a root, he/she can send events, such as the creation of accounts and sending coins to such accounts, among others. Such events are stored in a blockchain as logs. From the smart contracts created in the first stage (root creation), it is possible to consult any stored log by the root for audit purposes. We can also validate the data integrity guaranteed by the strength of the blockchain. This will allow to validate the integrity of the data, guaranteed by the strength of the blockchain.

With this, we complete our description of how all events generated by root are registered in the blockchain. These scenarios could be repeated with the creation of other types of users.

### 3.3. Assumptions and Security Requirements

Since building a security environment from scratch is complex, we make the following assumptions:The communication protocol: Considering the current world infrastructure where almost all applications technologies are mounted on the internet, we assume the TCP/IP protocol, where attackers might be present.Tunneling: Since messages transmitted over the internet can be manipulated by an attacker, we assume a tunneling secure communication using transport layer security (TLS), respectively, secure socket layer (SSL).Perfect cryptography: We assume that symmetric and asymmetric cryptography is not vulnerable to cryptanalysis.Collision free hash: We assume a perfect hashing function and so ignore the fact that hash functions cannot easily provide collision resistance.Secure blockchain and network communication: We assume that the blockchain network is strongly secure and reliable. That is, our mechanism disregards any possible attack in the blockchain mining process. We also assume that the communication with the blockchain is carried out through tunneling as above mentioned.Secure smart contract programming: We assume that the smart contract programming is not vulnerable.

The existing internet infrastructure already comes with mechanisms that reify some of these assumptions. So, our research meets the following requirements:The security protocol must guarantee that the root must be created correctly and securely in the AS. Although any user could become a root, the protocol guarantees that one and only one root is created in the system.The user requesting to be the root must have guarantees that the root creation was successful.The root’s smart contract must be created by a user account that has completed the protocol run without anomalies.All events (converted in logs) generated in the process of the root creation and produced subsequently must be stored in the blockchain through the root’s smart contracts.Only the root can execute transactions with its own smart contract in the blockchain. This also includes consulting the root’s logs. This means that no other user can execute transactions with the root’s smart contract.

The first three requirements rely on the security protocol run in the initial phase, while the rest rely on the logs in the blockchain.

## 4. First Stage: Root Creation Mechanism

First of all, we introduce the notation we use to describe a security protocol. Then, in the following subsections, we explain the first stage of *RootLogChain* by providing a general security protocol description; next, we describe the protocol, step by step. The last subsection explains the communication interaction between the protocol with the blockchain network.

### 4.1. General Notation

In symmetric cryptography, we use the same key to encode and decode a message; by convention, symbol ${\left\{\;\left|m\right|\;\right\}}_{K}$ is used to denote that a message m is ciphered under the key *K*. An agent is a computer process that uses client–server technology to establish network communication. To understand the communication notation of the protocol, see Table 1. The table describes a general Alice and Bob notation, which is used in the definition of our security protocol through communication events. A protocol is formed by two parts: the initial knowledge of the participants and a set of message steps. The initial knowledge of each agent is denoted as a list of messages. The steps consist of sending and receiving messages under the client–server technology; before sending a message, an agent can execute an operation that depends on a previously received message as part of a local process.

In Table 2, a short description of the notation used in the following sections is shown.

### 4.2. Protocol Specification

Following the experience reported in [41], and the design guidelines for security protocols to prevent replay and parallel session attacks [22], we propose the following security protocol: AS:[A,AS,K]A:[A,AS,K,D,Na=newNonce();]1.A→AS : getInitialNonce;A;NaKif(!root){  Nb=newNonce();  skiptostep2}else{skiptostep2′}2.AS→A : A;Na;NbKTo=hash(Na,Nb)skiptostep32′.AS→A : deny;A;NaKprotocolhasfinished3.A→AS : createRoot;D;ToKTo=hash(Na,Nb)if(!rootandvalid(To)){  LogP={A,AS,To,Hash(A,AS,To)}  M={LogP,D}  R=createSmartContracts(M)  ∃dx.dx∈D  skiptostep4}else{skiptostep4′}4.AS→A : R;To;dxKprotocolhasfinished4′.AS→A : deny;A;ToKAccepted

In general, the above protocol describes how any user acting as A could become a root, but if there already exists one in server AS, the protocol rejects such a request. When B accepts A, then A becomes a root, and it is also created in the blockchain employing smart contracts. Steps 1, 2, 3, and 4 denote a normal execution of the protocol without anomalies for any user wanting to become a root, while 2’ and 4’ are answers executed by AS rejecting the creation of the root because there exists one in the system or because the requirements are not met. The following subsection explains the protocol in detail.

### 4.3. Protocol Description

The explanation of the protocol comprehends the initial knowledge, normal execution of the protocol without anomalies, and possible executions of the protocol identifying anomalies.

#### 4.3.1. Initial Knowledge


AS:[A,AS,K]A:[A,AS,K,D]


The protocol assumes that we have a previous key distribution protocol, such as TLS or SSL. It means that any user, including an intruder, could establish tunneling with server AS. Therefore, at the beginning of the protocol, both A and AS know themselves, and they know the shared key, *K*, which was previously exchanged. All future messages are ciphered using this key. We remark that A knows D, where D is a set of data denoting root information, such as *username*, *password*, a *public key* of the user (wanting to become a root), a *gas* indicating the cost necessary to perform the transaction on the blockchain, and anything else important to be stored within the blockchain as an identity of the root.
A:Na=newNonce();

In addition, agent A at the start of the protocol creates a nonce $N_{a}$ as a challenge; it denotes a freshness property for the protocol. A nonce is a random, unrepeatable number of characters. Note that each instance of agent A should generate its nonce $N_{a}$.

#### 4.3.2. Steps without the Existence of a Root User


1.A→AS : getInitialNonce;A;NaKif(!root){  Nb=newNonce();  skiptostep2}2.AS→A : A;Na;NbKTo=hash(Na,Nb)skiptostep33.A→AS : createRoot;D;ToKTo=hash(Na,Nb)if(!rootandvalid(To)){  LogP={A,AS,To,Hash(A,AS,To)}  M={LogP,D}  R=createSmartContracts(M)  ∃dx.dx∈D  skiptostep4}4.AS→A : R;To;dxK


Step 1:Any user, acting as A, requests AS for a service called getInitialNonce and concatenates its identity A and the nonce $N_{a}$ previously created in the initial knowledge step, all ciphered with shared key *K*. When agent AS receives the message and decodes it, they assume that A is requesting the initial token because of the agent name A, and assume that $N_{a}$ is the challenge. AS checks that root is not yet created; then, they create a new nonce $N_{b}$, stores temporally these nonces and the requester A to identify the transaction. Note that expression !root denotes true when the root is not created yet.Step 2:Agent AS, using shared key *K*, ciphers the agent name A, nonce $N_{a}$ received in the previous step, and its new nonce $N_{b}$. Once received, agent A authenticates AS with nonce Na, accepts $N_{b}$ and forms the initial token by hashing the nonces created by A and AS ($\mathrm{To}=Hash(N_{a},N_{b})$), respectively. Agent A is ready to start step 3.Step 3:Agent A, ciphering the message with *K*, requests to create the root in the blockchain and attaches a set of data *D* denoting root information and the token. When AS receives the message and decodes it, they assume that A is requesting to create the root. AS checks if a root exists, verifies that the token received is valid by calculating the hashing of $N_{a}$ and $N_{b}$, and then creates the root in the blockchain, *R* being the answer obtained from the blockchain after having executed the remote function createSmartContracts(M). Then, AS chooses data $d_{x}$ included within *D* to respond to A.Step 4:AS, using *K*, ciphers *R* as evidence that a root is created, including the token linked to it and $d_{x}$. Agent A confirms the participation of AS with the token and accepts the root creation with *R* value. In this case, $d_{x}$ could be a hashed password or its email, for example.

#### 4.3.3. Steps Considering a Root User Existence


**First scenario: starting the protocol when the root has already been created.**

1.A→AS : getInitialNonce;A;NaKif(!root){  Nb=newNonce();  skiptostep2}else{skiptostep2′}2′.AS→A : deny;A;NaKprotocolhasfinished

Step 1:When agent AS receives the message and decodes it, they check whether the root has already been created; if it is true, then they skip to step $2^{\prime }$.Step 2’:Agent AS notifies agent A that such a request is denied by sending the string *deny*. AS also includes in the message the agent name of A and nonce $N_{a}$ to specify which nonce challenge and agent name were denied.



**Second scenario: any user can start the protocol correctly but cannot become a root user.**

3.A→AS : createRoot;D;ToKTo=hash(Na,Nb)if(!rootandvalid(To)){  LogP={A,AS,To,Hash(A,AS,To)}  M={LogP,D}  R=createSmartContracts(M)  ∃dx.dx∈D  skiptostep4}else{skiptostep4′}4′.AS→A : deny;A;ToKAccepted



Steps 1 and 2: We assume that agent A has executed correctly steps 1 and 2 and $\mathrm{To}=Hash(N_{a},N_{b})$ is valid. Another assumption is that To may not be valid.Step 3: When AS receives the message checks the request, they verify whether root has already been created; if it is true, then they immediately respond to A with step $4^{\prime }$. Another consideration is that although the root is not created yet, if the token is not valid, the answer is linked to step $4^{\prime }$.Step 4${}^{\prime }$: AS, using *K*, ciphers the notification that the root cannot be created with the token. Agent A receives the message and accepts the notification.

### 4.4. A Protocol Run and the Creation of the Root’S Smart Contract

Figure 3 illustrates the creation of the root without anomalies executing a protocol run; all steps are specified with white circled numbers within the green rectangle, which means that agent A, who is making the request, is who becomes the root. The figure illustrates three distributed entities: agent A, server AS, and the blockchain. The internal process of each entity is specified with its own rectangle. The two rectangles show examples of the variables exchanged in the protocol run. Note that in the figure, A only sends two different messages to be a root (steps 1 and 3); they also must accept messages from AS (steps 2 and 4). As mentioned in Section 3.3, the protocol assumes that the communication between AS and the blockchain is secured with a previous TLS/SSL protocol (pinpointed with gray tunneling in the figure).

To create a root, AS must follow some interactions with the blockchain. For example, the internal process, pinpointed with a circled 3a number in the figure, calls a function createSmartContracts(M), which is composed of the following:M={LogP,D}
where LogP is composed of {Op,A,AS,To,Hd}; Op is an event or operation type, usually an HTTP method (GET, POST, PUT or DELETE; in this particular case, it will be POST); and Hd is a hashed message to be stored in the blockchain (here, it is Hd=Hash(A,AS,To)). Let D be a set of data including the *username*, *email*, *password* and anything else important to execute the transaction within the blockchain (more details in Section 6.5). Remote function createSmartContracts(M) executes a transaction in the blockchain; specifically, the smart contract’s so-called *User* generates an instance which is the *Root* with information M. The transaction generates $R=\{A_{tr},A_{sc}\}$, which is a tuple of two elements: (a) a transaction address $A_{tr}$ denoting an identifier of the transaction of the smart contract; and (b) a smart contract address $A_{sc}$, which denotes an identifier within the blockchain. After AS receives R, it is concatenated with the token and $d_{x}$; in this case, $d_{x}$ is username. Then, it is ciphered using *K*, to be returned to A, who accepts the protocol run and becomes the root.

## 5. Second Stage: Root Events

Once the system has a root user, as explained in Section 4, we explain how all events generated subsequently are stored or consulted in the blockchain. Figure 4 illustrates the root events. On the left part, the communication is between the root and AS. On the right part, the communication is between AS and the blockchain. The events are triggered by the root, interpreted by AS, and stored or consulted in the blockchain; each includes a new token previously negotiated as an authentication procedure between the root and AS.

We have classified such events in two groups: (a) consulting logs for integrity issues; and (b) storing new events. Note that in the following subsections, we refer to the requester as the root.

### 5.1. Consulting Logs

In order to know how the log data are stored within the blockchain, AS has another service, *Consulting*; see the consulting process in Figure 4. Let C={R,To}. When the requester sends getCon(C), they hope to receive *L*, which is composed of the following tuple {To,log}; the first element is the token and the second element is log, which is received from the blockchain having executed findLog(R). log contains data sent by the root by its creation or by events generated; details about what a log contains are described in Section 6.3. In case C contains any information unable to be found in the blockchain, then log={"Error"}.

### 5.2. Storing Events

Figure 4 illustrates in the blue rectangle the logic of storing events. When root generates a new event (maybe because it is adding, updating, deleting, or consulting information via off chain), it involves generating a new log in the blockchain.This log is generated by executing addEvent(E) within server AS. Let assume that Root triggers an Event(E) operation by sending a message E, which is composed of the following:$$E=\{\mathrm{LogP},A_{sc},K_{p}^{+},\mathrm{Gas}\}$$

LogP is composed by {Op,A,AS,To,Hd} as explained in Section 4.4. Here, Op could be any HTTP method—GET, POST, PUT or DELETE; A denotes the source from where the service is being requested; AS denotes the target Audit Server; To is a token user identifier of such a specific transaction; Hd is a hashed message to be stored in the blockchain; $A_{sc}$ denotes the smart contract of the user; $K_{p}^{+}$ means the user who is requesting the event; and Gas denoting the cost necessary to perform this transaction on the blockchain.

Function event(E) calls a remote function addEvent(E), returning R, which is concatenated with token To forming C={R,To}, which is sent back to the Root. R, in this case, is similar to that explained in Section 4.4.

## 6. The Smart Contracts within the Blockchain

Blockchain involves a wide concept to understand. It involves mining, miners, peer-to-peer networks, how blocks are formed, etc. This section does not focus on the underlying blockchain technology; rather, we concentrate on the smart contracts that live within the blockchain.

### 6.1. Smart Contract Notation

We represent smart contracts as being similar to classes in object-oriented languages. They contain public (+) and private (-) attributes and methods; prefix (*) denotes an attribute with value that is auto-generated or internally calculated. A contract and its methods need to be called by a user address or another smart contract. Private methods can only be called within the contract, as long as public methods are still accessible from other contracts. Abstract methods are those without implementation. If a smart contract includes at least one abstract method, it is considered abstract and can only be instanced when all abstract methods are implemented.

Figure 5 illustrates the smart contracts as class diagrams. Constructors are shown in the figure with the same name as the contract. Some object-oriented characteristics of the smart contracts we have used are abstraction, inheritance, and dependency, as shown in the figure and explained in the following subsections.

### 6.2. Objectcontract

Similar to the Java programming language, where the *Object* class is the parent of all classes, we represent *ObjectContract* as the parent contract in our design. This contract contains two attributes (transaction address $A_{tr}$ and the contract address $A_{sc}$). Any contract that is created will have these two attributes. If any user wants to change something in the blockchain, they have to execute a *transaction*. As a result, it generates a new transaction address $A_{tr}$ on a contract $A_{sc}$. Contracts are referenced with their contract address. Hence, contract and transaction addresses can be obtained by methods getContractAddress() and getTransactionAddress(), respectively; the receipt of a transaction can be obtained by getReceipt().

### 6.3. Log

Each event generated in the system and received in the blockchain is registered by an internal smart contract called *Log*, inherited from *ObjectContract*; see Figure 5. A log in the blockchain is composed of those private attributes inherited from smart contract *ObjectContract*, and the following attributes:*Ablock: a block address in the blockchain.*idEvent: an event identifier. It is an autoincrement attribute identifying the current event.*Asig: a signature of the log.**when*: it is a timestamp T to know when the event has happened and when it was submitted to the blockchain.Op: to know the HTTP method (GET, POST, PUT or DELETE). POST is assigned automatically the first time the root is created.*where*: to know the source of the event*target*: to know the target of the event*token*:to know who has carried out the event for this specific event.Hd: hashed message required to describe more about the event.

All of these attributes are stored within the blockchain when method Emit(E) is called.

Note that the attributes marked with * are auto-generated within the smart contract; the rest are LogP and obtained from E. This is an abstract method, which we must implement in a specialized smart contract; in our case, this was called *Bitacora*. A particular log event can be obtained using getEvent(Atr), where $A_{tr}$ is the specific transaction address.

### 6.4. Bitacora

Each event in the system is registered by a smart contract called *Bitacora*. It inheritances all attributes and methods of internal smart contract *Log*, as you can see in Figure 5. The first time Bitacora is called, it creates its constructor Bitacora(M) (see Section 4.4 for more details about M), from M, information is extracted to form E and it calls Emit(E).

Emit(E) is a private method, inherited from *Log*, which stores the received values in the *Log*. This method can be called from constructor Bitacora(M) or Event(E).

setEvent(E) is another method, which can add events with information E by calling Emit(E) to be stored in the *Log*. In addition, all log events can be obtained using the method getEvents(), and we can obtain a particular log event by using the inherited method getEvent(Atr).

### 6.5. User: Root

A root user is created executing constructor User(M), which is as follows:M={A,AS,D,To}
M is a compound message, previously explained in Section 4.4; here, we crumble message D with more details:$$D=D_{1}::D_{2}$$
D is composed of two set of messages $D_{1}$ and $D_{2}$ respectively.
$$D_{1}=\left\{K_{p}^{+},gas,credentials\right\}$$
$D_{1}$ is formed by the user public key $K_{p}^{+}$ who will be the owner of the smart contract, and the key will be used to execute future transactions; *gas* denotes the cost necessary to perform the transaction on the blockchain; *credentials* is related with secret information of the root, such as *username*, *email*, and a hashed *password*. $D_{2}$ is used to add more information about the root and is required to be stored. It could be plain text or hashed data.

When the constructor of the smart contract *User* is executed, it creates its *Bitacora* and the first *Log* (calling the corresponding constructors). Smart contract *User* also generates the hashed message of all attributes, which are stored. These attributes can be accessed by its corresponding *getter* method as indicated in Figure 5. As a result of this transaction is generated R, which is as follows:$$R=\{A_{tr},A_{sc}\}$$

Smart contract address $A_{sc}$ denotes an identifier within the blockchain. It is required to find the instance of this smart contract within the ocean of the blockchain. Transaction address $A_{tr}$ denotes an identifier of the transaction within the smart contract. Note that each transaction in a smart contract produces a transaction address, which is required to find the transaction.

Smart contract *User* can add events, using method *addEvent(E)*; E is detailed in Section 5.2. This method calls method Event(E) of smart contract *Bitacora*. As a result of this transaction, another R is generated. In this case, the smart contract address $A_{sc}$ is the same as that sent within *e* since this method does not create a new smart contract.

In addition, to consult a bitacora and a specific log in the blockchain, this smart contract contains methods getBitacora(C) and getLog(C) respectively, where the following is true:$$C=\left\{\;\left|A_{tr},A_{sc},\mathrm{To}\right|\;\right\}$$

These methods call getEvents() and $getEvent(A_{tr})$ from smart contract *Bitacora*.

Figure 6 depicts how are created instances of *Bitacora* by triggering events by a user. The figure shows six events and all events are triggered by $K_{P}^{+}$ account; in the first event, a user using public key $K_{P}^{+}$ creates smart contract *Root*, with address contract $A_{sc}$; in this case, the public key account and the smart contract address belong to the created root; such an operation also creates *BitacoraR* with address $A_{B}$, storing $\left[L_{1}\right]$ in *Log1*, whose transaction address is $T_{R1}$. Events 1 to 5, the root, using $K_{P}^{+}$ adds new events (GET, PUT, POST and DELETE) to smart contract *Root*; in this case, *BitacoraR* adds $\left[L_{2},L_{3},L_{4},L_{5}\right]$ and creates the respective log for each event $T_{R2},T_{R3},T_{R4},T_{R5}$, respectively. The last event represents a consult operation (obtaining a log); there, the root wants to find a specific log C, which contains $T_{R2}$.

## 7. Prototype

To give a better idea about our mechanism, in this section, we provide a prototype available via https://git.io/JwAXa (accessed on 1 October 2021). This section describes how we have implemented the creation of the root with the protocol and its main proofs. We explain how Audit Server was developed, the technologies being implemented and how the smart contracts in the blockchain were generated.

### 7.1. Protocol Development

Figure 7 outlines the technologies used in the implementation of the security protocol introduced in Section 4.2, which can be downloaded from https://git.io/JwAXa (accessed on 1 October 2021). The figure shows three parts: the requester A, the server AS and the blockchain.

The requester part includes three different technologies that we used: Postman, Web, and Java. Postman was used for the developing phase; Web was used to prove that everything is interoperable and working correctly; and Java was used to test hundreds of threads trying to become the root.

Considering that our solution can be implemented as an extension in systems requiring data integrity evidence, we have found that the microservice architecture approach has become more popular in recent years. It was introduced as a solution to solve the monolithic problem, [42], developed as small, well-defined purpose and autonomous services deployed independently [43,44]. Microservice architecture offers various benefits, such as being small and focused, loosely coupled, language neutral, and having bounded context [44].

We have implemented AS following a software scheme as a service using a microservice architecture. The core base of the microservice (*Microservice Tech* in Figure 7) was based on Docker version 19.03.8 with Ubuntu 18.04 bionic core system and configured with web server NodeJS 10.15. We configured HTTPS to implement transport layer security (TLS) protocol with the port 443 on NodeJS. We used Docker as an open platform to administrate the services. Docker is known as a containers-as-a-service (*CaaS*) platform that uses a union file system (UFS) to deliver software in packages called containers. All containers are run by a single operating system kernel and therefore, use fewer resources than virtual machines [45,46].

In Figure 7, AS is composed of the API-Gateway and the Audit microservice. The API-Gateway is the main controller of the back-end. It receives operations requested by the requester, communicates internally with different services (in this case, with the Audit microservice), and emits a comeback answer. The API-Gateway follows the architecture of Gadge et al. [47].

The Audit microservice communicates directly with the blockchain. Library web3.js was used to connect with the blockchain. More details about the blockchain part are given in Section 7.2.

Figure 8 illustrates two complete runs of the protocol without anomalies: the upper part, using a web interface, and the bottom part, the messages exchanged using our interface implemented in Java language programming. In the upper part of the figure, A was our web implementation; all implemented functions, corresponding to A side, were programmed in JavaScript programming language. The microservices on the server side were programmed with NodeJS. The communication was ciphered with TLS protocol, shown in the figure as *https*. At the bottom of Figure 8, you can also see a run of the protocol executed with an interface we developed in Java programming language, and it uses *curl* program to execute client requests; there you can see the sender (-*n*–>) and the receiver (<–*n*-) messages, being *n* the step of the protocol.

### 7.2. Smart Contract Development

As you can see in Figure 7, the software component installed to execute the blockchain was Ganache CLI v6.4.3. The smart contracts were implemented in Solidity programming language (Solidity available via https://docs.soliditylang.org/en/v0.7.4/, accessed on 27 October 2021). The smart contracts can be downloaded from https://git.io/JRSFC (accessed on 27 October 2021).

Figure 9 depicts smart contract *User*, which can instantiate any user to become a root. To understand the complete syntax and semantic of Solidity can refer to https://solidity-es.readthedocs.io/es/latest/ (accessed on 27 October 2021). The smart contract shows at lines 5–9 the private attributes; lines 11–19 shows the Bitacora, which forms the set of logs of all events registered in the blockchain; lines 20 and 36 are private methods (for spaces issues we have hidden them); lines 49–62 depict the constructor, formed the first time when the smart contract is built, note that lines 53–57 validate automatically when the smart contract must be instantiated as a root; lines 63–74 shows how to add events in the smart contract, note that line 65 depicts how only the user who has created the smart contract can execute such a method; lines 60 and 72 store the Bitacora and form the logs; lines 75–84 are getters functions of the private attributes.

### 7.3. Execution Events and Costs of the Transactions

Figure 7 illustrates a microservice called *Audit* (on the AS side), which has two services: getLog and event(e). Their functionality is as described in Section 5.1 and Section 5.2 respectively. The next experiment reports the costs of the root creation and the four types of events carried out in the blockchain via the API-Gateway and the Audit microservice. These events are carried out after the root is created. The address values are as follows: $$K_{P}^{+}=0xAb8483F64d9C6d1EcF9b849Ae677dD3315835cb2$$
$$A_{sc}=0xd9145CCE52D386f254917e481eB44e9943F39138$$

Table 3 illustrates the cost for each transaction (last two columns) and also illustrates the events depicted graphically in Figure 6. Note that *idEvent* 1 denotes the creation of the contract, resulting the contract address $A_{sc}$, in this case, the *POST* event is deduced internally in the constructor of the smart contract. Events 2–5 are executions of method *addEvent* of smart contract $A_{sc}$ executed by $K_{P}^{+}$. All transaction costs are denominated in Wei. Wei refers to the smallest denomination of ether (ETH), the currency used on the Ethereum network. For instance, 1 ether (ETH) is equivalent to $1\times 10^{18}$ Wei.

Event 6 refers to a consulting process to know a specific log; because of that, such operation has no transaction and execution cost.

Figure 10 shows the result of consulting event 6. As can be seen, lines 13–19 of Figure 10 show the log data; they are matched with lines 12–18 of Figure 9. The rest of the fields shown in Figure 10 (after line 21) correspond to information related to the blockchain location.

## 8. Evaluation

This section provides two classes of validations carried out to our mechanism. The first one consisted of setting two environments where different types of agents, such as compliant and not compliant, are trying to become root users. The second is related to the latency and processing time of the transactions carried out by these types of agents.

### 8.1. Friendly and Hostile Environment

We designed two scenarios: a friendly and a hostile environment. The first one consists of a set of compliant agents running the protocol steps trying to become a root. The second one simulates an environment with non-compliant agents. Figure 11 illustrates, at the top, a compliant agent, two types of not compliant agents and server AS, the left part shows the steps of the protocol (described in Section 4.2) and at the center of the figure (pinpointed with coloured arrows) shows different traces executed by each type of agent.

The type of agents are described as follows:**Compliant agents:** they follow the protocol rules. In Figure 11 you can see a complete trace (black arrows) of a compliant agent becoming a root.**Non-compliant agents:** they do not follow the rules of the protocol and can send anything. We derived two sub-classes of these agents:
(a)Those configured to send anything but always starting the protocol normally (injection). You can see in Figure 11 how a non-compliant agent could also become a root (blue arrows), but if the server already created a root, it should reject its creation (red arrows); and(b)Those non-compliant agents that can start the protocol in any step of the protocol (injection). In Figure 11, you can see in green arrows this behavior. Note that this type of not compliant agents could start the run of the protocol at step 1 or 3 asynchronously; Figure 11 illustrates the rejection of such requests with red arrows.

To implement the behavior of Figure 11 we developed a tool with Java threads using an interface as shown in Figure 12. The tool can be configured with the number of requests, internally it creates N threads acting as A trying to become a root. Each request must send the following data *email, password, father and mother surname, name* and the target, in this case, the *server ip*. At the end of each run, the system must have a unique root in the blockchain, as shown in Figure 12 with the red rectangle. The figure also shows an example with 10 requests; 9 of them were rejected as the root (some of them with an underlined red mark).

With this application, we can carry out multiple proofs. At the beginning, we identified that some validations had been omitted in the implementation code related to the mutual exclusion problem with the critical procedure *root creation*, causing the duplication of roots (and consequently the duplication of smart contracts for the root). These problems were solved by implementing shared variable locks when a message of step 3 of the protocol arrived at AS. After repeating the proofs and making adjustments related to implementing the protocol as it was specified in Section 4.2, we created the only root. A version of the tool can be downloaded from GitHub: https://git.io/JRqCS (accessed on 27 October 2021).

### 8.2. Latency and Processing Time

Latency and processing time are important performance metrics when proving distributed applications because of the interactivity that an end-user can feel. In our proofs, latency is the time a client request (GET, POST, PUT or DELETE) takes to reach its destination and return. The processing time is the time a server takes to execute a request since it arrives until it is returned to the requester. The measurement related to these metrics are regarded in milliseconds (ms).

We have carried out some tests with the requests of the different types of agents introduced in the previous section. Table 4 shows a summary of one of the runs; there, the first column states ten agents numbered from 1 to 10, and their agent type is shown in the last column. The table is split in two: (1) the transaction latency from the client and (2) the processing time within the server. The *Start* and *End* columns are expressed in epoch time; and column *Time* expressed in miliseconds and it is the difference between *End* and *Start* times.

From the table, we can analyze that agent number 4 is the fastest transaction because the agent has not consumed time from the database and neither from the Blockchain, since being a non-compliant agent starts the protocol in the third step; hence, the server almost immediately rejects the request. On the other hand, agent number 6 is who becomes the Root and is the agent that takes the longest to complete the transaction. It is worth mentioning that this is the slowest transaction of all and involves processing in the off-chain database and within the Blockchain (although network mining time is not being considered due to the way Ganache CLI emulator works). In fact, out of registers from agents 4 and 6, the rest of the requests are very similar concerning server processing time, and their average processing time value is 1752 ms. This means 13,675 ms more time, approximately, when the transaction is executed on the Blockchain (this data could vary using another blockchain platform). The latency transaction time average on the network was approximately 3685 ms; it includes agents 4 and 6. The IP server AS was 54.87.22.33, and the transmission rate from the client was 1.76 Mbps upload and 17.97 Mbps download; the server characteristics were 4 GB RAM and 2.3 GHz 2VCPUs.

Blockchain implementations should provide the trade-off of the following characteristics: scalability, decentralization, latency, and security, as discussed in [34]. When an application is mounted on a public blockchain, we must take into account that each transaction generates high transaction costs, confirmed in Table 3. It also implies higher latency and processing time as reported in Table 4. *Scalability* is achieved in the proposed protocol by avoiding requests to the blockchain when an agent cannot become a root and reducing the transaction cost by sending hashed messages to the blockchain. In Figure 2, we show our *decentralized* cloud solution. In Figure 3 we show the interaction of at least three participant entities, one of them is a blockchain that itself represents a decentralized network. In Table 4, we show that our protocol shows a low *latency* speed when requests don’t proceed to become root user. Finally, we remark that we focus the *Security* of the proposed protocol on achieving the creation of a unique root user, as demonstrated in Section 8.1. Therefore, as we discussed, the trade-off is concerning to assure high security in the root creation, balancing the latency and decentralizing its design, and considering the scalability.

## 9. Conclusions and Future Work

In this paper, we showed the following: (1) stated a secure mechanism for the creation of the root in a distributed environment that allows to record and secure the transactions generated by the root events through smart contracts using blockchain; (2) provided a mechanism for auditing issues in the initialization phase of a system guaranteed by blockchain’s strengths concerning integrity and authenticity; and (3) perform how to obtain the traceability of the transactions logs stored in a blockchain.

The mechanism consists of two stages: the first one is a security protocol that permits the creation of a *root* user in a distributed system; the event of the root creation is stored in a blockchain, through an *Audit Server*. The second stage catches all events generated by the root, which are then stored in the blockchain through the smart contract generated in the first stage. With the smart contract, we can consult and track any log previously stored by the root.

*RootLogChain* can be adapted as an extension of systems requiring an audit characteristic because it was thought to be embedded in architectures based on microservices. So, we have adopted a microservice architecture in the back-end part with the aim that it can be adapted to other systems. One of the microservices connects with the front-end and the other with the blockchain network to store the events generated by the root.

For validating the functionality of our prototype, we have executed numerous security tests to guarantee the creation of a unique root. We have developed a scenario simulating compliant and non-compliant agents. For this, we have built a tool with java threads to simulate various clients trying to become a root. During our experiments, we identified many good programming practices that programmers often overlook (as we did) from the protocol specification, which could result in duplicating the root creation, leading to an incorrect implementation of the protocol (as we pointed out in the evaluation section.)

As shown in this research, we must consider that a distributed application mounted on a public blockchain induces high transaction costs due to a longer latency and processing time. So, the trade-off is to ensure high security in root creation while balancing efficiency. We must prioritize decentralized design and consider scalability.

This is ongoing research and we are planning to report other research results. Although here we have already reported some statistics regarding latency and processing time, we are also considering to report the performance of transactions in terms of memory use and energy expenses when it involves off-chain versus blockchain. In Section 3.1, we clearly framed two general phases and narrowed down the scope of this research in the first—the initialization phase of a distributed system. However, we are already working on the deployment phase (the second) and we plan to report our research progress in future work. Additionally, our current implementations were carried out on the Ethereum platform, so we believe that a comparison with others, such as Hyperledger Fabric, could be interesting for performance comparison.

## Figures and Tables

**Figure 1 sensors-21-07669-f001:**
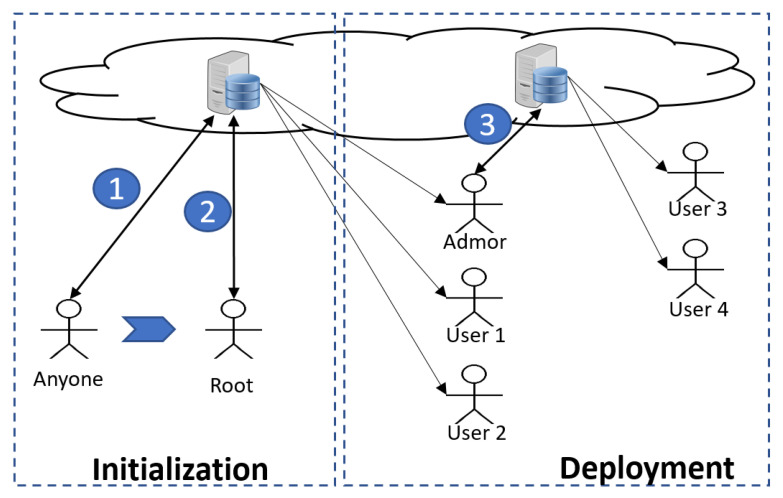
General phases for the generation of users in computer systems based on the cloud; circle numbers denote steps.

**Figure 2 sensors-21-07669-f002:**
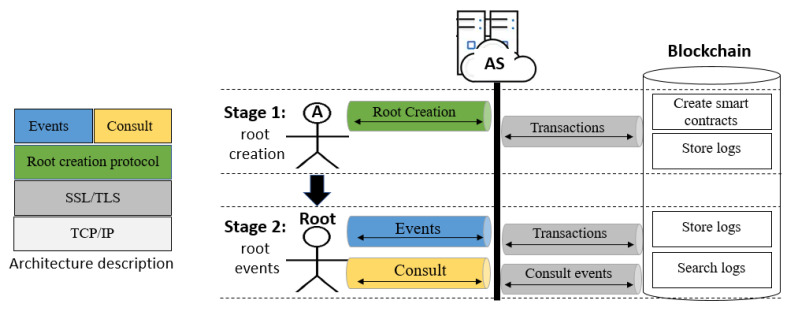
Architecture protocol and two stages played by four entities: agent A, the *Root*, AS and the Blockchain.

**Figure 3 sensors-21-07669-f003:**
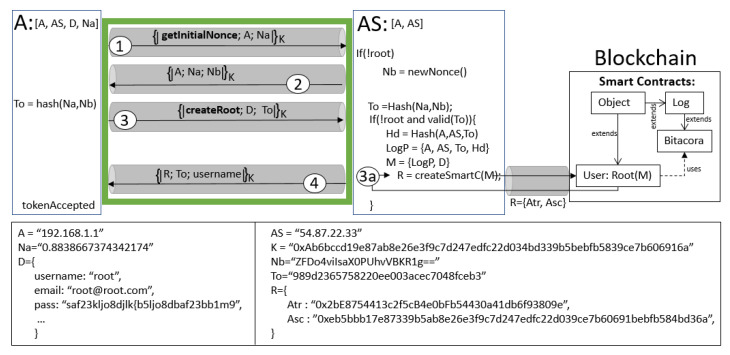
Root creation: a run of the protocol without anomalies and the interaction with the blockchain.

**Figure 4 sensors-21-07669-f004:**
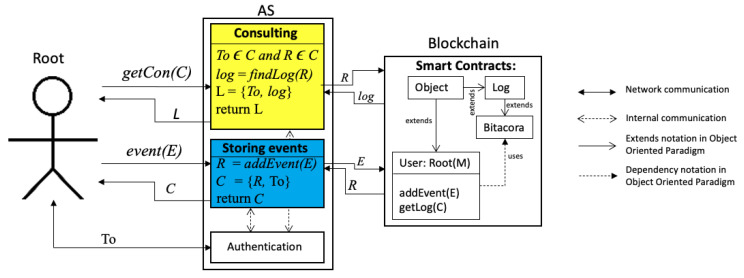
Consulting and storing logs.

**Figure 5 sensors-21-07669-f005:**
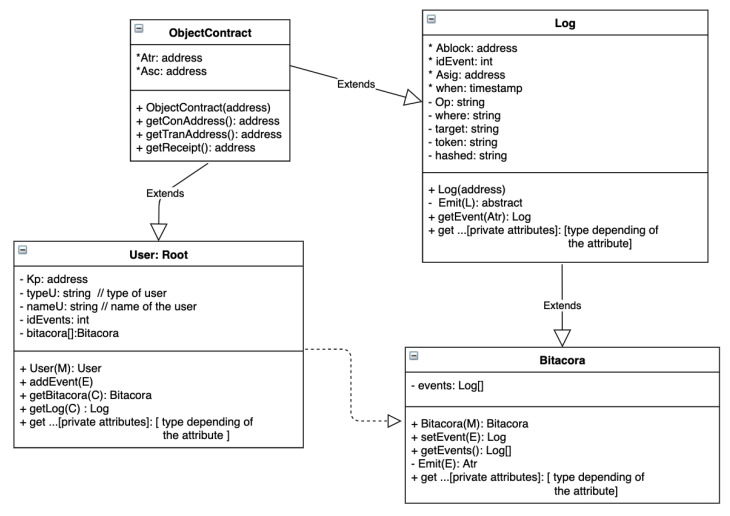
Smart contracts illustrated as class diagrams.

**Figure 6 sensors-21-07669-f006:**
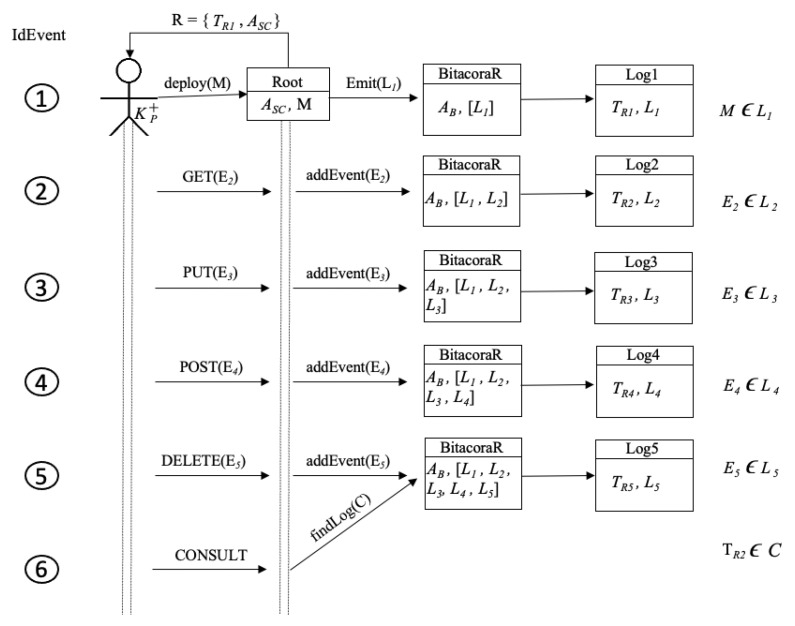
Events’ tree.

**Figure 7 sensors-21-07669-f007:**
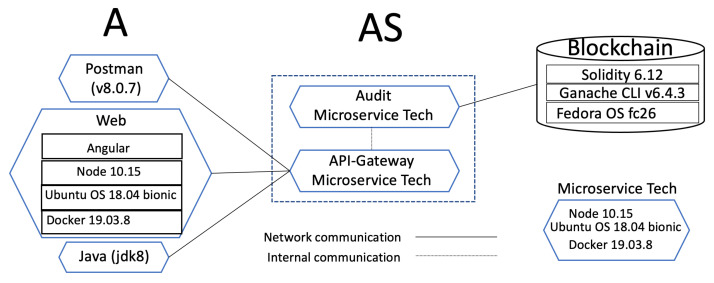
Technologies implemented in the protocol.

**Figure 8 sensors-21-07669-f008:**
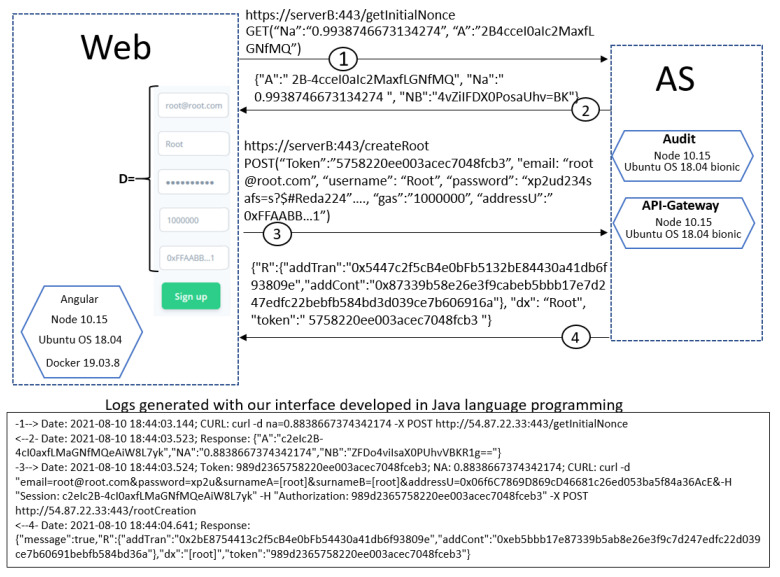
The complete protocol without anomalies.

**Figure 9 sensors-21-07669-f009:**
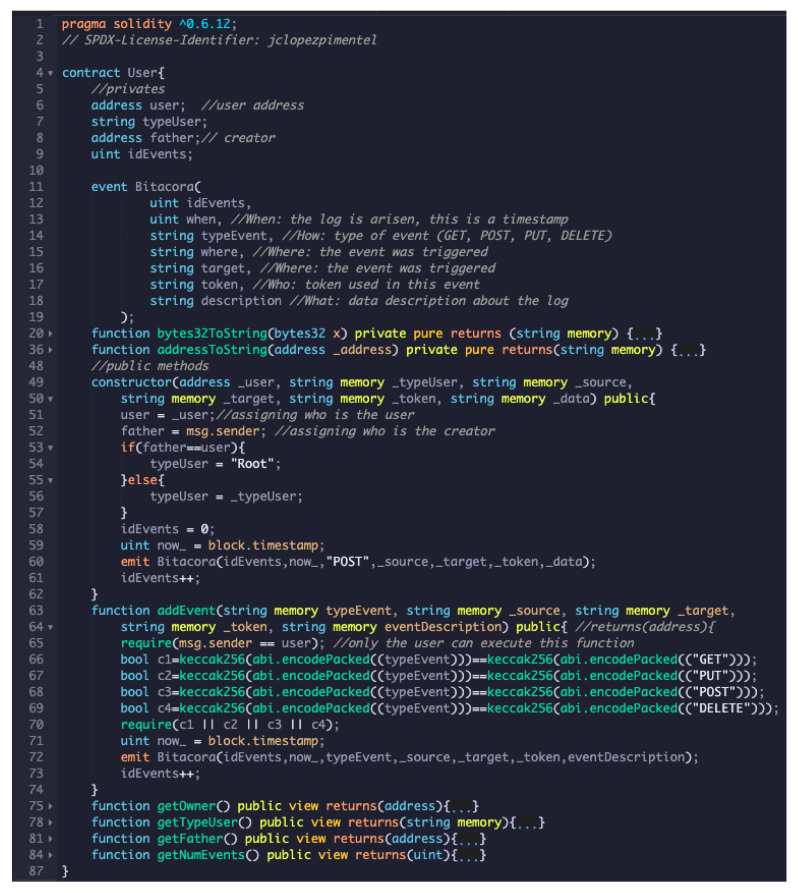
User smart contract in Solidity language programming.

**Figure 10 sensors-21-07669-f010:**
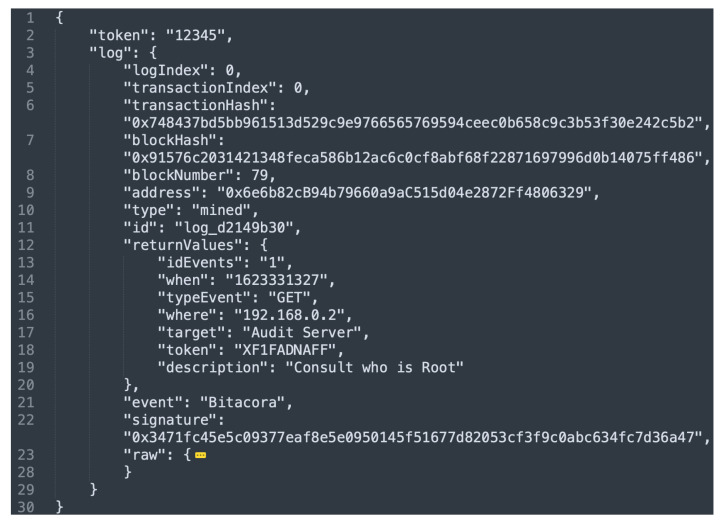
Log obtained from the blockchain.

**Figure 11 sensors-21-07669-f011:**
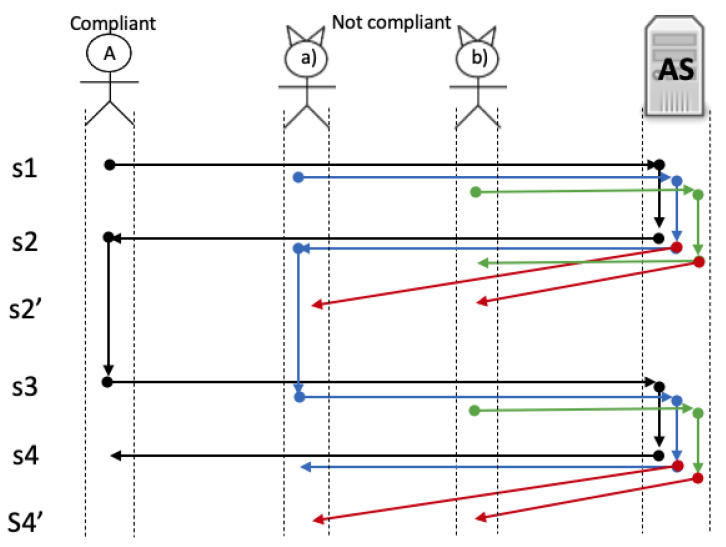
Behavior of compliant and not-compliant agents.

**Figure 12 sensors-21-07669-f012:**
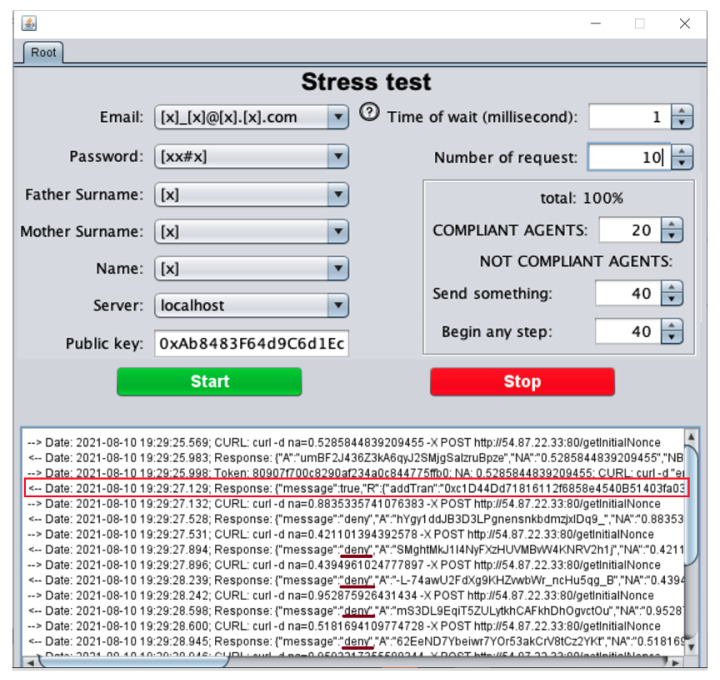
User interface to test the creation of the root.

**Table 1 sensors-21-07669-t001:** Conventions of a security protocol.

Abbreviation	Description
Initial knowledge:	
$A:[ik_{1},ik_{2}]$	Initially agent *A* knows $ik_{1}$ and $ik_{2}$.
Message sending and receiving:	
n.A→B:m	At step *n* agent *A* sends message m to agent *B*,
	which *B* receives.
Local process by *B*:	Between steps *n* and n2, *B* calculates $m_{2}$ from
$m_{2}=f\left(m\right)$	function f(m) as a local process.
$n2.B\to A:m_{2}$	At step n2 agent *B* responds to *A* with message
	$m_{2}$, which *A* receives.

**Table 2 sensors-21-07669-t002:** Notation: short description.

Abbreviation	Description
A	Sender user wanting to become a root
*B*	Receiver or target
AS	Audit Server
$N_{a}$	Nonce, denoting a random non-guessable number
*K*	A symmetric key
m	Any general message
${\left\{\;\left|m\right|\;\right\}}_{K}$	A cipher message, where m is encrypted with key *K*
*To*	Token
Hash(m)	Hash function with m as a parameter
Hd	A hashed message to be stored in the blockchain, e.g., Hd=Hash(m)
Op	Event or operation type, usually an HTTP method: GET, POST, PUT or DELETE.
LogP	Message composed by {Op,A,AS,To,Hd}
D	A set of personal data, such as Gas, username, pass, etc.
M	Message with information to create the root composed as
	M={LogP,D}
T	Timestamp
$A_{tr}$	Transaction address
$A_{sc}$	Smart contract address
E	Event message composed as $\left\{\mathrm{LogP},A_{sc},K_{p}^{+},\mathrm{Gas}\right\}$
R	Blockchain transaction answer compound as $\left\{A_{tr},A_{sc}\right\}$
C	Answer to the root after triggering an event composed as {To,R}
$K_{p}^{+}$	Public key address
Gas	The cost necessary to perform a transaction on the blockchain
L	Answer message sent to the root after a consulting process, composed as {To,log}
log	Answer message received from the blockchain after a finding log consulting process, composed as {R,Ablock,Asig,idEvent,LogP}

**Table 3 sensors-21-07669-t003:** Cost of the transactions of the events generated in Figure 6.

Executor	Contract	Method	Internally Calculated			Passed as Parameter					Cost	
			IdEvent	When	Event	Owner	Source	Token	Data	Type Event	Transaction	Execution
$K_{P}^{+}$		Constructor	1	1623331326	POST	$K_{P}^{+}$	192.168.0.1	XF12ADN2FF	New Root created		1,101,920	72,2440
$K_{P}^{+}$	$A_{sc}$	addEvent	2	1623331327		$K_{P}^{+}$	192.168.0.2	XF1FADNAFF	Consult who is Root	GET	43,090	16,954
$K_{P}^{+}$	$A_{sc}$	addEvent	3	1623331329		$K_{P}^{+}$	192.168.0.2	XF1FADNAGF	Updating Root Info	PUT	43,031	16,959
$K_{P}^{+}$	$A_{sc}$	addEvent	4	1623331331		$K_{P}^{+}$	192.168.0.3	XF1FADNBGF	Admin user was created.	POST	43,356	16,964
$K_{P}^{+}$	$A_{sc}$	addEvent	5	1623331332		$K_{P}^{+}$	192.168.0.3	XFGFGDNBGF	Admin user was deleted.	DELETE	43,492	16,972
$K_{P}^{+}$	$A_{sc}$	getLog(*C*)	6	1623331339		$K_{P}^{+}$	192.168.0.1	XFGFGDNACG				

**Table 4 sensors-21-07669-t004:** Transaction latency from the client and processing time in the back-end.

Agent Number	Latency A (Client)			Processing Time in the Server AS			Agent Type
	Start	End	Time (ms)	Start	End	Time (ms)	
1	1633651876243	1633651883000	6757	1633651882402	1633651884241	1839	Not-compliant sends something
2	1633651880123	1633651883900	3777	1633651882968	1633651884773	1805	Compliant
3	1633651876234	1633651883000	6766	1633651882640	1633651884419	1779	Not-compliant sends something
4	1633656003180	1633656005576	2396	1633651882758	1633651882764	6	Not-compliant begins any step
5	1633651876864	1633651883476	6612	1633651882399	1633651884004	1605	Not-compliant sends something
6	1633651876720	1633651896753	20,033	1633651882385	1633651897813	15,428	Compliant (Root)
7	1633651880065	1633651883663	3598	1633651883070	1633651884860	1790	Not-compliant sends something
8	1633651877244	1633651883863	6619	1633651882724	1633651884553	1829	Not-compliant sends something
9	1633651877141	1633651883332	6191	1633651882401	1633651884015	1614	Not-compliant sends something
10	1633651880769	1633651884332	3563	1633651883165	1633651884927	1762	Not-compliant sends something

## Data Availability

Some implementations of this research are available online. The security protocol implementation can be downloaded from https://git.io/JwAXa (accessed on 27 October 2021). The smart contracts can be downloaded from https://git.io/JRSFC (accessed on 27 October 2021). A version of the tool used for the stress proofs can be downloaded from GitHub: https://git.io/JRqCS (accessed on 27 October 2021).

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
