# Peer review of "RootLogChain: Registering Log-Events in a Blockchain for Audit Issues from the Creation of the Root"

_sensors, 2021, doi:10.3390/s21227669_

Round 1

Reviewer 1 Report

The authors explore a blockchain-based mechanism to address the issues of fairness and transparency in system logs. The proposed methodology enables an end-user to audit the logs and the immutability of the blockchain ensures that the logs are not tampered with (not even by the root user). The manuscript outlines the mechanism involved to initialize the logs in a distributed manner and security analysis of possible attacks on the system. The experiments confirm the behavior of the proposed methodology under compliance and non-compliance scenarios.

I have the following concerns with the manuscript:

  1. The authors frame the initialization phase of a system for root creation as one of the stages of the proposal. The issue with the claim is that there is hardly any motivation regarding the same in the introduction. Also, the blockchain system essentially solves this exact issue of deciding who adds a block to the blockchain in a distributed environment. The consensus algorithm solves the issue mentioned by the authors. 
  2. The next phase uses the blockchain to achieve transparency and auditability. The application of blockchain is straightforward and does not present any novelty. 

The lack of motivation and the selected tool's feature solving the issue makes the proposal weak. 

Regarding the language of the manuscript, I suggest the authors get editing help from someone with full professional proficiency in English.

Author Response

Response to Reviewer 1 Comments

Font code to understand the main changes in the article:

Changes: highlighted red text 

We appreciate your comments.

Point 1:  The authors frame the initialization phase of a system for root creation as one of the stages of the proposal. The issue with the claim is that there is hardly any motivation regarding the same in the introduction. Also, the blockchain system essentially solves this exact issue of deciding who adds a block to the blockchain in a distributed environment. The consensus algorithm solves the issue mentioned by the authors.

Response: Thanks for this suggestion. We modified some paragraphs in the Introduction section (highlighted in red font) to emphasize the motivation of this work.  In particular, the third paragraph emphasizes the problem.

Besides, a couple of parenthetical paragraphs of the Introduction section were moved to Section 2, “Background and Related Work”. Such paragraphs could divert attention to the main idea of the Introduction, and they corresponded more with related work. 

Point 2:  The next phase uses the blockchain to achieve transparency and auditability. The application of blockchain is straightforward and does not present any novelty.

Response: Thanks for this comment. We agree with the first argument of this comment because our contribution is not directly on the blockchain network. On the other hand, we can say that the article's contributions are emphasized in the Introduction section (page 2, itemized with bullets). In fact, the paper also distinguishes the strength that blockchain already provides (page 2, at the end of paragraph 2) and the aim of our research (at the beginning of paragraph 2, page 2). 

We focus our main novelty on dealing with the audit characteristic of a system with a strong traceability characteristic that blockchain already provides, instantiating it with a critical user as the root.

Point 3: The lack of motivation and the selected tool's feature solving the issue makes the proposal weak.

Response: Thanks for this suggestion. We describe the motivation issue in Point 1. In addition, on Page 2, paragraph 3 has been paraphrased to emphasize the technologies used; also, in the Conclusion section (paragraphs 3 and 4, highlighted with red font) has been abstracted the technologies applied in sections 7 and 8.

Point 4: Regarding the language of the manuscript, I suggest the authors get editing help from someone with full professional proficiency in English.

Response: Thanks for this suggestion. We have proofread the manuscript and corrected the grammatical errors throughout the document.

Additionally: 

  • We have updated and added 11 new recent references throughout the paper [3, 4, 5, 6, 7, 8, 28, 34, 35, 36, 37].
  • We have paraphrased the last paragraph of the conclusion to clarify our future work plan and be more explicit (highlighted with red font). 
  • We have added a new sub-section 8.2 titled “Latency and processing time.” Besides, we include a new table 4 describing the latency and processing time proofs carried out with ten agents from both the front-end and back-end. The sub-section discusses the data of the table and some statistics derived from it. We think that this new section improves the paper giving more support in the evaluation proofs.  

Reviewer 2 Report

First of all, List of References includes some publication which are published before 10-20 years (see [3, 5, 9, 10, 14, etc.]) – I think that the relevant of the discussed topic is not supported by them. The relevance of the research should be well confirmed as such articles do not help for this

The first section “Introduction” is quite long and generally covers the essence of the second section on related work. The positive is that the last paragraphs indicate the important for such a section - contribution and structure of the article.

In general, the article is well structured and presents the research information in an appropriate way. However, it is noteworthy that references are used in sections that are expected to be the author's work - for example section 7 – see part 7.1 and links in part 7.2. It is desirable to specify the connection of this reference with the author's research.

In general, the last section does not clearly show what future work is planned.

Author Response

Response to Reviewer 2 Comments

Font code to understand the main changes in the article:

Changes: highlighted red text 

We appreciate your comments.

Point 1: First of all, List of References includes some publication which are published before 10-20 years (see [3, 5, 9, 10, 14, etc.]) – I think that the relevant of the discussed topic is not supported by them. The relevance of the research should be well confirmed as such articles do not help for this

Response: Thanks for this suggestion. We have updated and added 11 new recent references throughout the paper [3, 4, 5, 6, 7, 8, 28, 34, 35, 36, 37].

Point 2: The first section “Introduction” is quite long and generally covers the essence of the second section on related work. The positive is that the last paragraphs indicate the important for such a section - contribution and structure of the article. 

Response: Thanks for this suggestion. A couple of parenthetical paragraphs of the Introduction section were moved to Section 2, “Background and Related Work”. Such paragraphs could divert attention to the main idea of the Introduction, and they corresponded more with related work. Also, we modified some paragraphs in the Introduction section (highlighted in red font) to emphasize the motivation, problem description, objective, and validations carried out in this work.  

Point 3: In general, the article is well structured and presents the research information in an appropriate way. However, it is noteworthy that references are used in sections that are expected to be the author's work - for example section 7 – see part 7.1 and links in part 7.2. It is desirable to specify the connection of this reference with the author's research. 

Response: Thanks for this suggestion. Originally, the introduction part of section 7 had two paragraphs; taking into account your suggestion, we have deleted the first paragraph and paraphrased the second one. In addition, we have rewritten some paragraphs in sections 7.1 and 7.2 to distinguish our research implementation from the citations to other works (highlighted with red font). 

Point 4: In general, the last section does not clearly show what future work is planned.  

Response: We have paraphrased the last paragraph of the conclusion to clarify our future work plan and be more explicit (highlighted with red font). 

Additionally: 

  • We have added a new sub-section 8.2 titled “Latency and processing time.” Besides, we include a new table 4 that describes the latency and processing time proofs carried out with ten agents from both the front-end and back-end. The sub-section discusses the data of the table and some statistics derived from it. We think that this new section improves the paper giving more support in the evaluation proofs.  
  • We have proofread the manuscript and corrected some grammatical errors throughout the document.

Reviewer 3 Report

The paper is overall well written and presents an interesting study. The following concerns may be addressed:

  1. The literature review is not exhaustive and should include more recent papers such as the following and other similar papers related to blockchain: 10.1109/TII.2020.3007817, 10.1109/JIOT.2020.3002711,  10.1109/TLA.2020.9387645.
  2. Do a proper proof read.
  3. Please discuss the four way trade-off of blockchains in relation to the proposed scheme.
  4. Provide latency results for the proposed scheme.

Author Response

Response to Reviewer 3 Comments

Font code to understand the main changes in the article:

Changes: highlighted red text 

We appreciate your comments.

Point 1: The literature review is not exhaustive and should include more recent papers such as the following and other similar papers related to blockchain: 10.1109/TII.2020.3007817, 10.1109/JIOT.2020.3002711,  10.1109/TLA.2020.9387645.

Response: Thanks for this suggestion. We have updated and added 11 new recent references throughout the paper  [3, 4, 5, 6, 7, 8, 28, 34, 35, 36, 37]. 

Point 2: Do a proper proof read.

Response: Thanks for this comment. We have proofread the manuscript and corrected the grammatical errors throughout the document.

Point 3: Please discuss the four way trade-off of blockchains in relation to the proposed scheme.

Response: Thanks for this suggestion. We have included at the end of section 8.2 a paragraph discussing the four-way trade-off: scalability, decentralization, latency, and security; contrasting it with our proposal and the strength of blockchain. As we will explain in Point 4, section 8.2 is a new sub-section included as a suggestion of such a comment.

Point 4: Provide latency results for the proposed scheme.

Response: Thanks for this suggestion. We have added a new sub-section 8.2 titled “Latency and processing time.” Besides, we include a new table 4 that describes the latency and processing time proofs carried out with ten agents from both the front-end and back-end. The sub-section discusses the data of the table and some statistics derived from it. We think that this new section improves the paper giving more support in the evaluation proofs. 

Additionally: 

Some paragraphs have been edited in the Introduction section (highlighted in red font) to emphasize the motivation, problem description, objective, and validations carried out in this work. 
